# Observationally quantified reconnection providing a viable mechanism for active region coronal heating

Kai E. Yang [1,2,3], Dana W. Longcope[2], M.D. Ding[1,3] & Yang Guo [1,3]

The heating of the Sun's corona has been explained by several different mechanisms including wave dissipation and magnetic reconnection. While both have been shown capable of supplying the requisite power, neither has been used in a quantitative model of observations fed by measured inputs. Here we show that impulsive reconnection is capable of producing an active region corona agreeing both qualitatively and quantitatively with extreme-ultraviolet observations. We calculate the heating power proportional to the velocity difference between magnetic footpoints and the photospheric plasma, called the non-ideal velocity. The length scale of flux elements reconnected in the corona is found to be around 160 km. The differential emission measure of the model corona agrees with that derived using multi-wavelength images. Synthesized extreme-ultraviolet images resemble observations both in their loop-dominated appearance and their intensity histograms. This work provides compelling evidence that impulsive reconnection events are a viable mechanism for heating the corona.

[1] School of Astronomy and Space Science, Nanjing University, Nanjing 210023, China. [2] Physics Department, Montana State University, Bozeman, MT 59717, USA. [3] Key Laboratory for Modern Astronomy and Astrophysics (Nanjing University), Ministry of Education, Nanjing 210023, China. Correspondence and requests for materials should be addressed to M.D.D. (email: dmd@nju.edu.cn) or to Y.G. (email: guoyang@nju.edu.cn)

Heating of the Sun's corona is often attributed to either Alfvén waves[1,2] or nanoflares[3,4] of which many aspects have been studied at length[5–11]. Recent investigations have yielded new insights into the coronal heating mechanism. For example, observations from the Atmospheric Imaging Assembly (AIA)[12] on board Solar Dynamics Observatory (SDO) have shown that the total energy flux observed in low-frequency Alfvén waves is sufficient to supply the energy heating the quiet corona, but not the active corona[13]. This does not, however, rule out the possibility of waves heating the active corona, if observations had underestimated the actual energy flux, perhaps occurring on numerous randomly distributed loops[14]. Moreover, there is certainly some energy in the frequency range outside that observed. In order to account for coronal heating, the energy flux in Alfvén waves would have to be dissipated. Several energy dissipation mechanisms have been proposed, including resonant absorption[15–17] and phase mixing[18], in an inhomogeneous plasma. Nevertheless, it is not yet clear what fraction of the energy flux carried by Alfvén waves can be dissipated in the corona by any of the mechanisms proposed.

The alternative scenario, that of magnetic reconnection, assumes that the corona is heated by numerous small-scale energy release events called nanoflares[3,4]. This hypothesis is supported by the reasonable correspondence between the differential emission measure (DEM) observed and that predicted from random heating by nanoflares[19]. Another investigation showed that the corona could be well simulated using the observed solar velocity spectrum and Ohmic dissipation from an artificially high resistivity[20].

Magnetic reconnection occurs when an electric field, directed parallel to the local magnetic field, changes the connectivity of field lines, by allowing them to move independently of the plasma itself. Such a process is able to release energy stored in the large-scale magnetic field. A key measure of independent motion is the apparent slippage of field line footpoints relative to the plasma in which they would otherwise be anchored[21–23]. It is possible to observe and measure this non-ideal motion by tracing field lines from one footpoint to its conjugate footpoint in a sequence of coronal field models. If the starting footpoint is fixed to move with the plasma, reconnection will cause the conjugate footpoint to move at a velocity different from the plasma. This velocity difference, which we hereafter call the non-ideal velocity, is proportional to the parallel component of the electric field integrated along that field line, a measure of the reconnection rate[21]. The non-ideal velocity is also known as the slipping velocity found during solar flares[23–25]. If reconnection is somehow heating the corona, as nanoflare models assume, then the local heating rate will be proportional to the reconnection rate and thus to the non-ideal velocity. Our observational measure of the reconnection electric field provides a heating rate without assuming a particular dissipation mechanism, such as Ohmic heating used in many investigations[20,26,27], about which there is still great uncertainty[28]. Here, we show that this hypothesis leads to an equilibrium, active region corona qualitatively and quantitatively similar to observations.

## Results

**Energy released by magnetic reconnection.** Reconnection releases magnetic energy only if it occurs in the presence of current. Transferring a finite amount of flux, $\delta\Phi$, across a net current $I$, will release energy $\delta E = I\delta\Phi$[29]. This is the electromagnetic work done by the reconnection, and is valid regardless of how that released energy is converted into non-magnetic forms. If this flux element is reconnected in impulses repeating with a mean interval of $\tau_r$, the average heating power will be

$P_i = I\delta\Phi/\tau_r$. This expression accounts for the integrated reconnection electric field through Faraday's law, $\delta\Phi/\tau_r \sim -\int \mathscr{E}_{\|}\mathrm{d}s$. The flux transfer event will slip the loop's footpoints a distance roughly equal to the diameter of the reconnected flux element, $\mathscr{L} = \sqrt{\delta\Phi/\overline{B}}$, where $\overline{B}$ is the mean field strength at the photosphere where the non-ideal motion is observed. The mean non-ideal velocity will then be $v_s = \mathscr{L}/\tau_r$. The parallel current across which this reconnection occurs is $I = \alpha\delta\Phi/\mu_0$, where $\alpha$ is the local twist in the force-free field: $\nabla \times \mathbf{B} = \alpha\mathbf{B}$. The average heating rate for the single flux element is therefore $P_i = \alpha v_s \mathscr{L}\delta\Phi\overline{B}/\mu_0$. The flux elements may be too small to resolve, so a resolvable photospheric area $A$ will include $A\overline{B}/\delta\Phi$ sub-resolution elements. The mean energy flux, $F$, input into the coronal volume anchored to that area will be

$$F = \frac{\sum_i P_i}{A} = \frac{1}{\mu_0}\alpha\mathscr{L}v_s\overline{B}^2. \qquad (1)$$

This is the rate of heating due to energy released by repeatedly reconnecting flux elements of diameter $\mathscr{L}$ independent of the mechanism by which the energy is eventually dissipated. There is not yet an ab initio theory of magnetic reconnection predicting the size of elemental reconnection events. With the improvement of high-resolution instruments[30–32], some details of the magnetic strands have been observed[9,33]. The observed width of the magnetic strands might or might not be directly related to the diameter of a reconnected tube. Nevertheless, for simplicity, we take a value of 160 km for the parameter $\mathscr{L}$ in the model and assume it is the same for all flux elements. Compared with recent observations[9,33], this value would be regarded as the upper limit for the width of the magnetic strands.

**Measuring non-ideal velocity.** The non-ideal velocity of any field line is measured with the following procedure (Fig. 1a). We reconstruct the coronal magnetic field from a non-linear force-free field model[34] and through it trace field lines from positive to negative footpoints, denoted p and n respectively. We perform this for magnetic equilibria from two closely spaced times, $t_0$ to $t_1$ separated by $\delta t = t_1 - t_0 = 720$ s, the cadence of the vector magnetograms from the Helioseismic and Magnetic Imager (HMI)[35,36]. At the initial time $t_0$, we trace a field line from $\mathbf{p}_0$ to $\mathbf{n}_0$, indicated by the yellow loop. The plasma elements initially located at those points move according to the photospheric velocity field derived using the Differential Affine Velocity Estimator for Vector Magnetograms (DAVE4VM)[37] applied to the same pair of HMI vector magnetograms. By time $t_1$ this flow has taken $\mathbf{p}_0$ to $\mathbf{p}_1$ and $\mathbf{n}_0$ to $\mathbf{n}'_1$. Had the corona evolved without reconnection, $\mathbf{n}'_1$ would be conjugate to $\mathbf{p}_1$ through the coronal field found at time $t_1$. Owing to the presence of reconnection this is not the case and the footpoint conjugate to $\mathbf{p}_1$ is located at some other point $\mathbf{n}_1$. The difference in these locations, $\delta n = |\mathbf{n}_1 - \mathbf{n}'_1|$, is therefore due entirely to the reconnection electric field[21]. The non-ideal velocity $v_s = \delta n/\delta t$ measures the integrated reconnection electric field along that one field line. To obtain the corresponding velocity of the positive footpoint, we fix the negative footpoint. We would expect the two measures to yield the same value of heat flux since the electric field integral would be the same. In practice, the heat flux related to the two results would differ slightly due to differences in the actual field line used, but must converge as $\delta t \to 0$.

**Modeling the corona.** The next step is to determine the plasma's response to the heat input derived above[5]. There has been extensive work modeling the corona in one and more

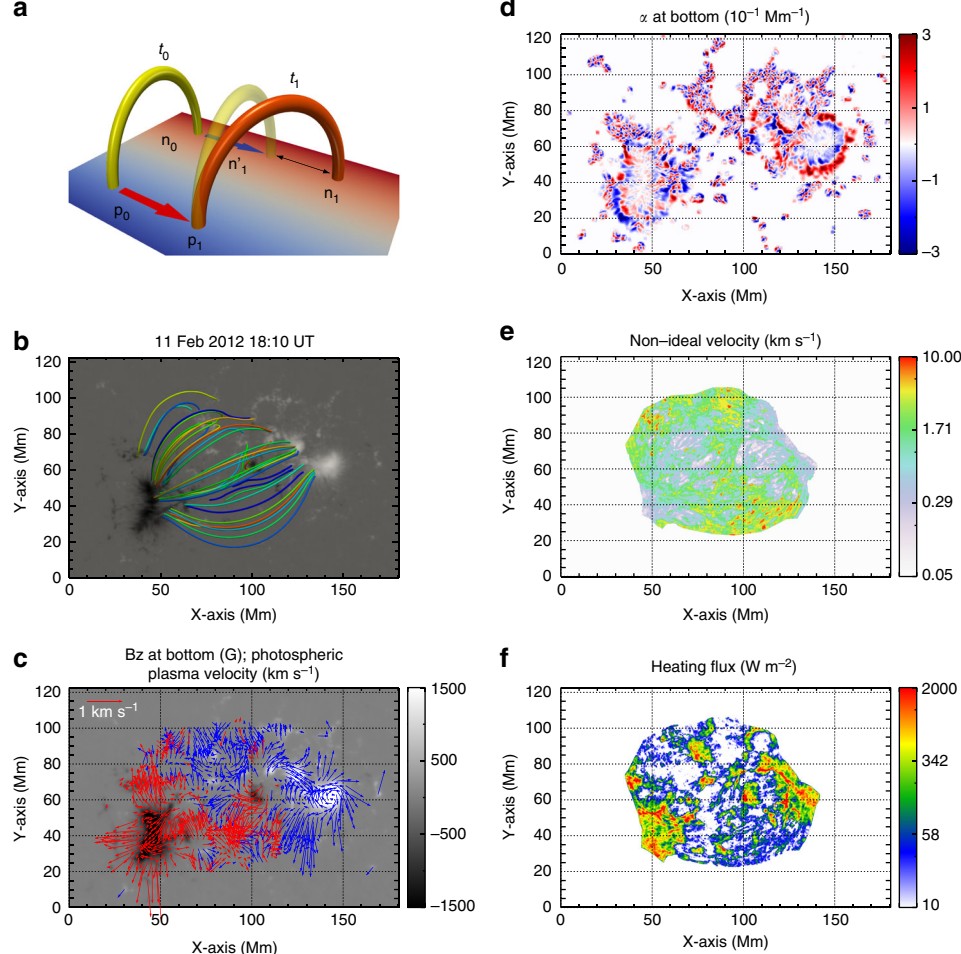

**Fig. 1** A model showing the non-ideal velocity and the measurement results. **a** Coronal loops show the measurement of the non-ideal velocity. The letters **p** and **n** stand for the conjugate footpoints of a loop and the subscripts 0 and 1 refer to time labels for the start and end time, $t_0$ and $t_1$, for one measurement, respectively. The yellow loop represents the initial loop at $t_0$ and the transparent yellow loop is the hypothetical version at time $t_1$ under the assumption of ideal MHD. The footpoints are advected to new places by the photospheric plasma flow, as indicated by the red and blue arrows in positive and negative polarities, respectively. The orange loop is the actual coronal loop at time $t_1$. The distance denoted by the two-headed arrow is the non-ideal distance $\delta n$ due to magnetic diffusion (reconnection). **b** The extrapolated magnetic field at 18:10 UT. **c** The photospheric plasma velocity field overplotted on the vertical magnetogram from HMI (background). The blue and red arrows represent the plasma flow with positive and negative magnetic flux, respectively. **d** The force-free parameter, $\alpha$, calculated from the vector magnetogram at 18:10 UT. **e** The measured non-ideal velocity. Here, we only calculate the non-ideal velocity where both footpoints of the field lines are rooted within the field of view and $|\mathbf{B}| > 20$ G in the magnetograms. **f** The heating flux calculated from the reconnection model with the parameter $\mathscr{L} = 160$ km

dimensions[38–57]. Our modest objective, however, is simply to obtain the distribution of density and temperature from a specified heat input. Toward this end we assume coronal equilibrium and obtain a value of temperature and density at each point along an equilibrium loop[38–41,58] (details in Methods section). To justify our equilibrium assumption, we note that the reconnection event frequency for a typical non-ideal velocity is 5 km s⁻¹/160 km = 0.03 Hz. Since this rate is high compared to radiative cooling rate[59], impulsive reconnection will have the effect of a steady input, known as a nanoflare storm. Though there are a lot of dynamic processes in the real solar corona, the equilibrium approximation is still a good one under conditions such as these described above.

We trace the field line at $t_1$ from a given coronal point to its two footpoints. We then average the reconnection heat flux from the footpints, $F_p$ and $F_n$, which are evaluated by Eq. 1 at those points in the photosphere. The volumetric heating function used in the equilibrium model depends on the dissipation mechanism, about which we have made no assumption. We follow previous

authors[39,58] by adopting an exponential heating distribution $H(s) = H_0 \exp(-s/\mathscr{L})$, where $s$ is the distance from the nearest footpoint to the initial coronal point. We determine $H_0$ using the energy into the loop averaged over that from the two footpoints $2\int_0^{\frac{L}{2}} H(s)\mathrm{d}s = (F_p + F_n)/2$, where $L$ is the total loop length. We express the heating scale length $\mathscr{L} = \mathscr{R}L/2$, where $\mathscr{R}$ is a free parameter in our model. The density and temperature for the specified coronal point are taken to be those from the corresponding loop solution. This procedure is then repeated for every point in the corona serving as the initial point for a new loop. Our method resembles those of some previous studies[41,60], but we populate every coronal point independently rather than superposing distinct loops.

**Application to observations**. We perform the above computation on the active region (AR) NOAA 11416 on 11 February 2012, which was well observed by SDO. We expect this region to be well approximated by our equilibrium assumption because the magnetic flux variation was less than 1% during 2 h and no obvious

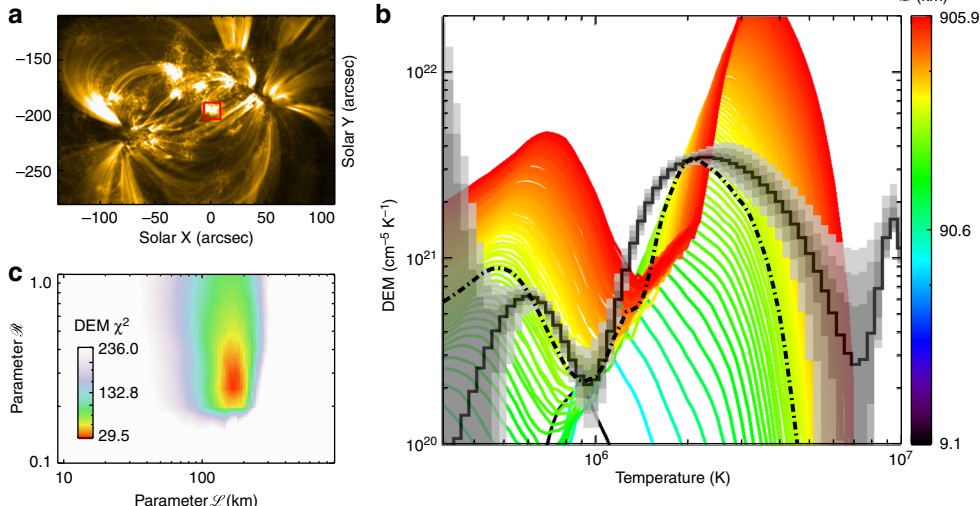

**Fig. 2** DEM distribution from the model and the discrepancy $\chi^2$ between the model and observations. **a** The image at EUV 171 Å observed by AIA. The red box indicates the area where the DEM is calculated and shown in **b**. **b** The DEM distribution calculated from the model by fixing $\mathscr{R} = 0.3$ but varying the scale length of the cross section of the loop, $\mathscr{L}$ (see the color bar). The solid black curve is for the DEM inferred from the AIA observations at six wavelengths averaged over 12 min. The gray areas with different transparencies represent the results with $1\sigma$, $2\sigma$ and $3\sigma$ from the Monte-Carlo test. The dash-dotted line is the curve from the model with $\mathscr{L} = 160$ km. **c** The discrepancy $\chi^2$ between the modeled DEM and the observed one plotted as a color scale over parameter space

flares occurred during the time of our modeling. The non-ideal velocity, twist parameter $\alpha$ and the heating flux are calculated using an HMI vector magnetogram pair from 17:58 and 18:10 UT. Figure 1b–d shows the reconstructed magnetic field at 18:10 UT, the photospheric plasma velocity computed using DAVE4VM, and the distribution of $\alpha$ at the photosphere. Figure 1e, f shows the magnitude of the non-ideal velocity and the heating flux $F$, found from Eq. 1, respectively.

For the sensitivity analysis of the parameter $\mathscr{L}$, and searching for the best fitting of the parameter $\mathscr{R}$, we vary the values of these two parameters over a range (9 km < $\mathscr{L}$ < 900 km and 0.1 < $\mathscr{R}$ < 1.0) and compute the density and temperature throughout the corona and from these synthesize a column DEM (Fig. 2b and Supplementary Fig. 1b) over a subarea (Fig. 2a). This is then compared with the DEM inverted directly from multi-channel AIA observations in the same subarea. This yields a discrepancy quantified by $\chi^2$. The results show that 160 km is a good choice of $\mathscr{L}$, and the optimal free parameter of $\mathscr{R}$ is approximately 0.3 (Fig. 2c). We also check the sensitivity of $\mathscr{L}$ and the fitting quality of $\mathscr{R}$ by comparing the intensity histograms formed over a larger subarea (shown in Fig. 3a) in six different AIA bandpasses (Fig. 3b–g and Supplementary Fig. 2b–g). The result is similar to that from the DEM distribution (Fig. 3h).

We use this optimized parameter, $\mathscr{R} = 0.3$ from DEM, to synthesize extreme-ultraviolet (EUV) images of the entire AR, and compare these to SDO/AIA images in Fig. 4. Many corresponding structures can be found between them, e.g., the brightening loops, moss structures and large loops (indicated by numbers 1–4 in Fig. 4). The similarities are remarkably good for a model with only one global free parameter, $\mathscr{R}$, although the agreement is not perfect. Note that the non-ideal velocity is structured at very small scales (Fig. 1e). This structuring is mapped to the heat flux $F$ (Fig. 1f) leading to the appearance of isolated loops in the synthetic EUV images (Fig. 4). This is a notable point of agreement considering, as stated above, the image was constructed voxel by voxel, and not from superposing elemental loop structures.

Comparison with the previous studies[59,61] shows the value of $\mathscr{L}$ to lie within the range of the characteristic size of the magnetic

strands. In particular, in the recent high-resolution observations, some ultra-fine channels were found with a diameter of 100 km and co-spatial with brightenings in EUV bandpasses[33]. Thus, the choice of our parameters seems very reasonable. It could be further constrained by the future instruments such as Daniel K. Inouye Solar Telescope.

## Discussion

To further analyze the reliability of our method, we deduce the relative errors in the key parameters, i.e., the standard deviation from the mean, by performing 50 new versions of the entire calculation after adding random errors with a standard deviation of 20 G to the vector magnetogram at the lower boundary. The results, shown in Fig. 5, demonstrate that the relative errors decrease with the mean values for the plasma velocity, non-ideal velocity and heating flux. Those pixels with magnetic field strength greater than 100 G have relative errors less than 0.6 (Fig. 5d–f). The heat flux averaged over the strong magnetic field region (|**B**| > 100 G) is approximately 800 W m$^{-2}$ comparable to that known to heat a relatively weak active region[62]. Thus, the energy released by nanoflares can be directly estimated by our method, yielding a quantitative and spatially distributed heating rate, without being extrapolated from the occurrence distribution of larger flares[63].

The DEM is a promising diagnostic tool when we analyze the multi-wavelength coronal emissions. The practice of its measurement does, however, have limitations. To probe these limitations we re-compute the DEM using the synthesized EUV images from the modeled corona. We compare this with the DEM computed from the model and that computed directly from the observations (Fig. 6). We can see that the DEM from the synthesized EUV images (the red line in Fig. 6) is very close to that from the model with the largest departure occurring at lower temperatures. This suggests that, at least in the higher temperature domain, the DEM inversion can yield reasonable results.

Even adopting an optimization method, there remains a discrepancy in the DEMs at the highest temperatures. This may result from our use of an equilibrium loop to estimate the plasma response to heating. We have demonstrated above that the mean

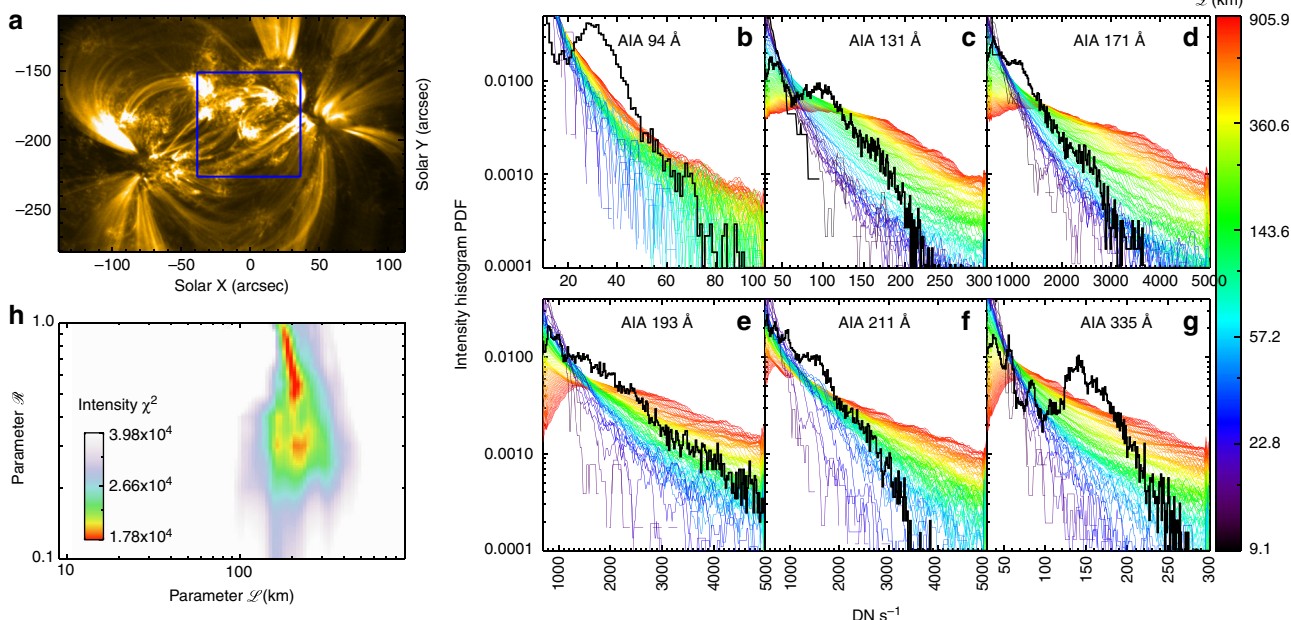

**Fig. 3** Intensity histogram from the model and the discrepancy $\chi^2$ between the model and observations. **a** The image at EUV 171 Å observed by AIA. The blue box refers to the area over which the discrepancy is computed between the intensity histograms from the model and observations as shown in **b**–**g**. **b**–**g** The intensity histogram calculated from the model by fixing $\mathcal{R} = 0.3$ but varying the scale length of the cross-section of the loop, $\mathcal{L}$ (see the color bar). The solid black curve stands for the intensity histogram from the AIA observations at six wavelengths averaged over 12 min. **h** The sum of the Pearson's $\chi^2$ of the intensity histogram between model and observations over all of the six wavelengths, $\chi^2 = \sum_i \chi_i^2$, where $i$ is the index of the six AIA channels. It is plotted as a color scale over parameter space

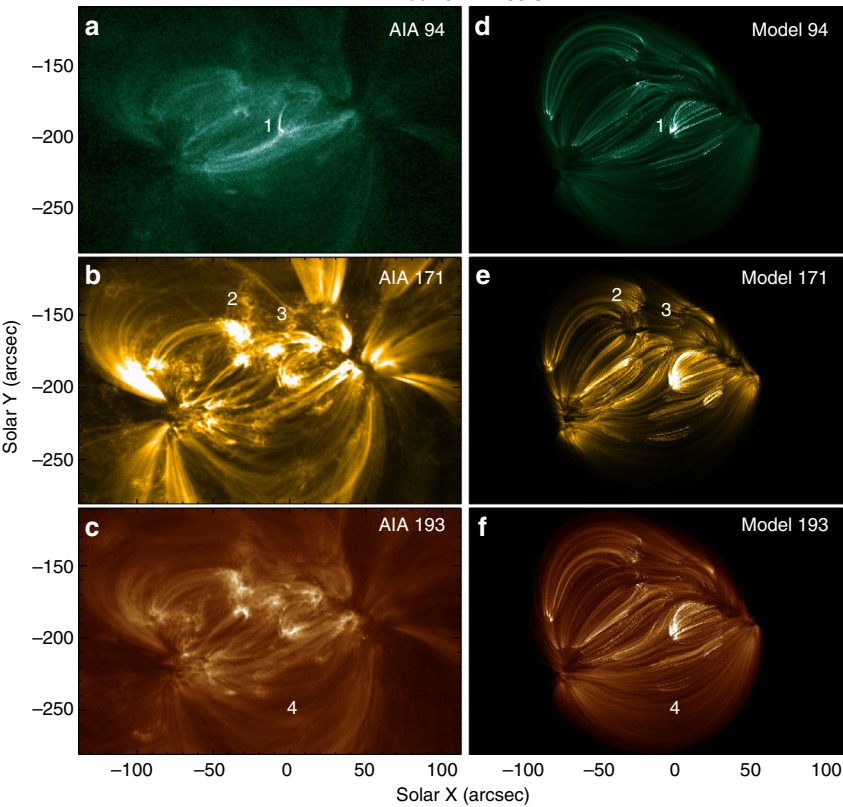

**Fig. 4** Observations and synthesized EUV images showing the loop structures. The left column (**a**–**c**) shows the AIA observations at three channels. The right column (**d**–**f**) shows the synthetic images at the same channels generated with $\mathcal{L} = 160$ km and $\mathcal{R} = 0.3$. All images use linear color scales with units of DN s$^{-1}$, and corresponding pairs use the same color scale. The numbers indicate the corresponding structures, including brightening loops (1), moss structures (2 and 3) and large loops (4), respectively

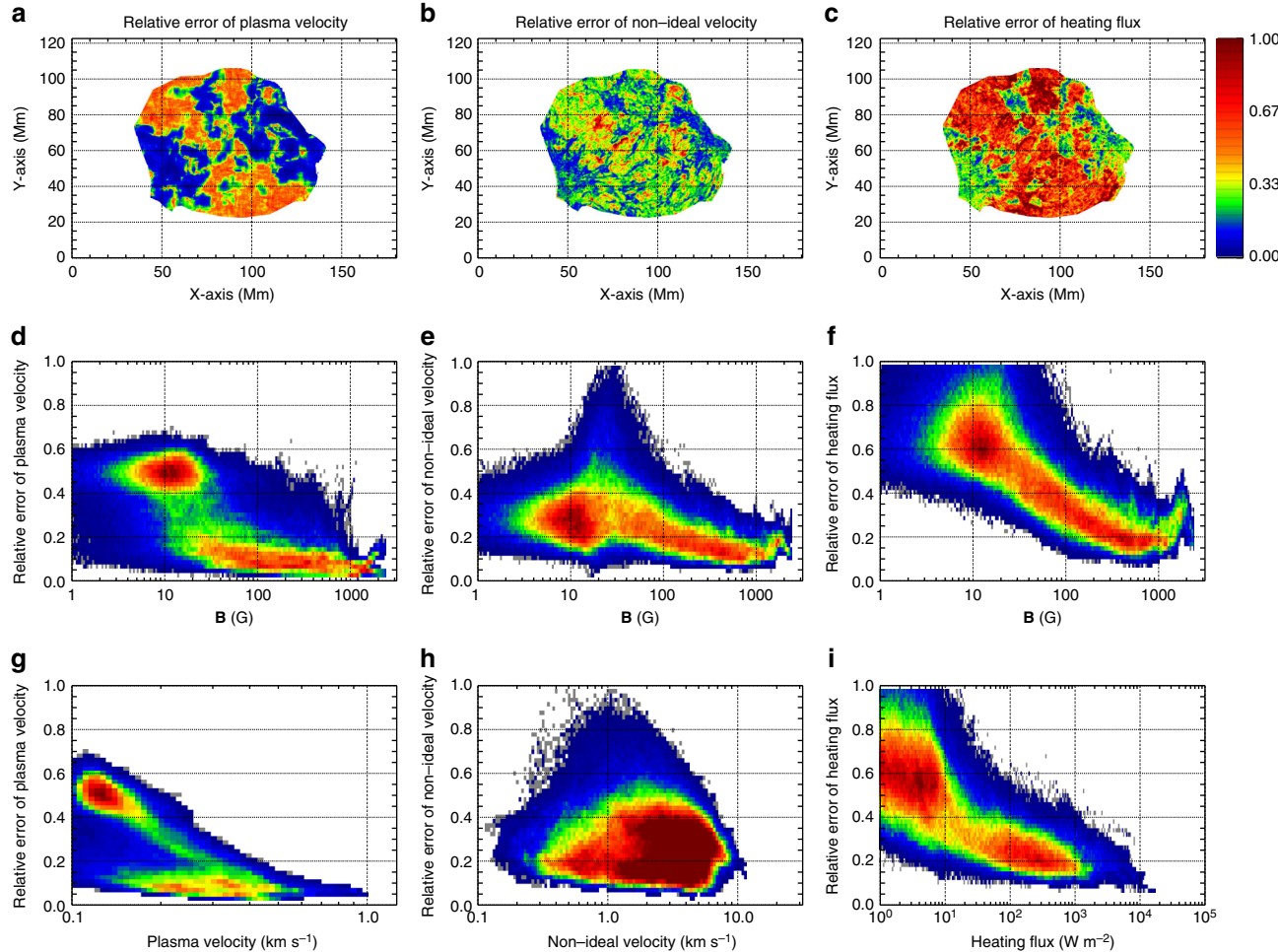

**Fig. 5** Relative errors of the plasma velocity, non-ideal velocity and heating flux. The errors are estimated as the standard deviation over the mean value from 50 Monte-Carlo simulations. **a–c** The spatial distributions of the relative errors of the three parameters. **d–f** The histogram density distributions of the errors in dependence on the magnitude of the magnetic field. **g–i** The histogram density distributions of the errors in dependence on the mean values of the corresponding parameters

time between typical impulsive events is short enough to justify the equilibrium assumption. There may, however, exist significantly larger events occurring at a significantly lower frequency, which would fall outside the equilibrium assumption. Larger, less frequent nanoflares are, in fact, known to produce locally high temperature and high density in the corona[5]. Another questionable assumption was that the loop had an upright, semi-circular axis geometry and a uniform cross-sectional area. In fact, violation of these assumptions might lead to some discrepancy between the model and observations. The axis geometry will primarily affect the loop's legs, where the scale height is the smallest. Any inclination away from the purely radial legs, as we have assumed, would therefore presumably enhance the DEM only at the lowest temperatures.

Variation in the cross-sectional area, on the other hand, would be inversely proportional to the magnetic field of the loop. In most cases, the loop constricts from the corona to the chromosphere gradually, and such a constriction occurs most significantly at the loop's feet. This diminished area, and thus diminished volume, would decrease the DEM at the lowest temperatures[64]. We therefore expect that accounting for these effects would produce results akin to that of using a different value of the parameter $\mathscr{R}$ in the present model[58].

In conclusion, we have developed a reconnection-based model which can estimate the heating rate from the observed non-ideal velocity. The model can predict the temperature and density distributions of the corona, at least to first approximation, with only one global free parameter. Our model avoids using an artificially high resistivity, or specifying any form of dissipation at all. The predicted thermal structure of the corona, in particular the DEM and intensity distributions, resemble the observations both qualitatively and quantitatively. Thus, our study indicates that magnetic reconnection is a plausible heating mechanism to maintain an active region corona remarkably similar to the observed one.

## Methods

**Coronal magnetic field and photospheric plasma velocity.** We use HMI level 1.5 vector magnetograms from the Space Weather HMI Active Region Patches data[65] for AR 11416 from 17:58 and 18:10 UT on 11 February 2012, and the pair of three-dimensional magnetic fields is modeled with the non-linear force-free assumption by using the optimization method[34], which minimizes a functional combining the magnetic field divergence, the Lorentz force, and the error in the observations. The lateral and top boundaries are set according to the method presented in a previous study[34]. The photospheric plasma velocity is inferred by solving the magnetic induction equation using the DAVE4VM[37] and the window size used for it is selected as 23 pixels.

**Equilibrium loop.** The coronal plasma density and temperature are found by assuming the magnetic loop to be in equilibrium[38,39,58], with heating balanced by

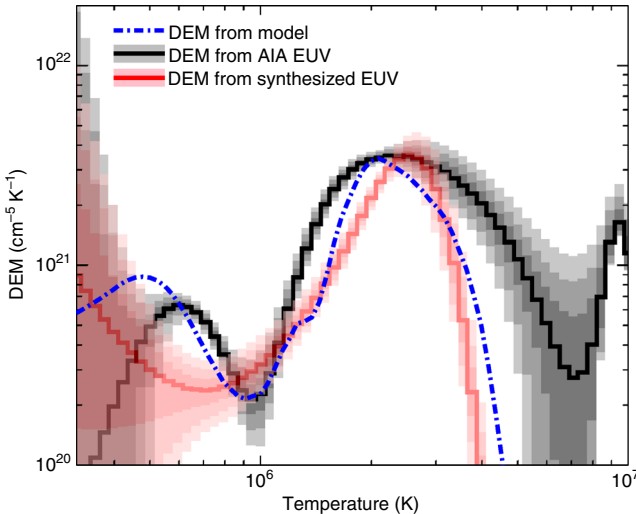

**Fig. 6** Comparison of DEM profiles constructed from the model, inverted from observations and from synthesized EUV images. The black solid line and the gray areas indicate the DEM profile inverted from the AIA observations, which is the same as that in Fig. 2. The red solid line and pink areas indicate the DEM profile inverted from the synthesized EUV images for the same area as that of the black solid line. The blue dashed line is the DEM profile from the model

the radiation and thermal conduction. We solve the energy equation

$$-\frac{P^2}{4k_B^2 T^2}\Lambda(T) + \frac{\partial}{\partial s}\left(\kappa\frac{\partial T}{\partial s}\right) + H(s) = 0, \tag{2}$$

where $s$ is the length along the loop starting from one footpoint, $k_B$ is the Boltzmann constant, $T$ and $P$ are the temperature and gas pressure, respectively. The first term is the energy loss by radiation, where $\Lambda(T)$ is the radiative loss function, which we take from CHIANTI v8.1[66,67] complemented with the coronal abundance determined by Schmelz et al.[68]. The second term is the thermal conduction, where $\kappa = \kappa_0 T^{2.5}$ is the Spitzer conductivity, and $\kappa_0 = 10^{-6}$ erg cm$^{-1}$ s$^{-1}$ K$^{-7/2}$. The third term is the local volumetric function normalized to yield the heat flux given by Eq. 1. We take an exponential profile for $H(s)$ as described in the section 'Modeling the corona'. The boundary condition is set as,

$$\begin{cases} T(0) = 10^4\,\text{K}, \\ \kappa\frac{\partial T}{\partial s}\big|_{s=0} = 0. \end{cases}$$

We solve for the pressure using hydrostatic balance

$$\frac{dP(s)}{ds} = -\frac{g_\odot \overline{m}}{k_B T(s)} P(s)\cos\left(\pi\frac{s}{L}\right), \tag{3}$$

where $\overline{m}$ is the average particle mass, $g_\odot$ denotes the gravitational acceleration on the photosphere and the cosine comes from assuming an upright semi-circular geometry, which is an approximation of the real geometry from the three-dimensional coronal magnetic field. Thus, the effects of the inclined loop on the profiles of temperature and density are neglected in our calculation.

The above equations are solved as an initial value problem starting from $s = 0$. For each value of $L$, $F$ and $\mathcal{R}$, we perform the initial value integration using the initial condition $P(0)$ as a parameter adjusted via the shooting method[69] to satisfy the condition of symmetry about the loop top, $\partial T/\partial s|_{s=L/2} = 0$. The density and temperature are recorded at a series of points along the solution. In this way we create a set of equilibrium loop solutions characterized by different values of length $L$, heat flux $F$ and ratio of heat scale length $\mathcal{R}$. We perform a single synthesis for a fixed value of $\mathcal{R}$ by tracing all field lines as described in the section 'Modeling the corona'. For each field line we determine $F$ and $L$, and then interpolate from the loop set described above to deduce the temperature and density at the coronal point in question.

**Differential emission measure**. The column DEM derived from our model is calculated directly by $\text{DEM}(T) = \text{d}(n(T)^2 h)/\text{d}T$, where $h$ is the line-of-sight distance and $n(T)$ is the plasma density with temperature $T$. Then the synthetic fluxes of the optically thin EUV images are obtained by $I_i = \int \text{DEM}(T)K_i(T)\text{d}T$, where $K_i(T)$ is the response function of the AIA instrument for the $i$th wavelength from SDO package in Solar Software. This produces synthetic images with units of DN s$^{-1}$, exactly the same as the observations; no scaling is performed. On the other hand,

the DEM can be inverted from observations using the regularized inversion method[70] with the AIA EUV images at six bandpasses (94, 131, 171, 193, 211 and 335 Å) in the temperature range of $5.5 < \log_{10}(T) < 7.0$. We conduct 10,000 Monte-Carlo realizations in order to estimate the errors in the results.

**Data availability**. The data that support the findings of this study are available from the corresponding author upon reasonable request.

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

## Acknowledgements

We thank Eric R. Priest and James A. Klimchuk for valuable discussions. SDO is a mission of NASA's Living With a Star Program. K.E.Y., M.D.D., and Y.G. are supported by NSFC under grants 11733003, 11773016, 11703012, and 11533005, and NKBRSF under grant 2014CB744203, and the China Scholarship Council 201606190132.

## Author contributions

K.E.Y. analyzed the observational data and performed model calculations. D.W.L. contributed to the theoretical formulation of the model. M.D.D. conceived the study and supervised the project. Y.G. jointly supervised the project. All authors discussed the results and wrote the manuscript.

## Additional information

**Competing interests:** The authors declare no competing financial interests.

