## [Peer Review File · Nature Communications]

Reviewers' comments:

Reviewer #1 (Remarks to the Author):

On "Observationally Quantified Reconnection Providing a Viable Mechanism for Active Region Coronal Heating" By Yang et al.

What are the major claims of the paper?

The manuscript claims to demonstrate that impulsive reconnection events (nanoflares) are a viable candidate for heating the corona. Further, that matching gross diagnostics of plasma temperature distribution are sufficient to make the primary claim.

General: Unfortunately I do not find the work contained in the manuscript terribly novel. Also, I do not think that the outcomes are unique, largely as a result of the model-observation metric applied. As a result it is very difficult to assess if the rather obvious claim being made in the title and abstract is indeed the case. I will provide some examples below, but cannot see how anything other than a radically different approach could be used to back the claim made above.

Are they novel and will they be of interest to others in the community and the wider field?

I [opinion] think it is widely believed that magnetic reconnection at small-scales (nanoflares; as discussed originally by Parker) is important for the transport of mass and energy throughout the outer solar atmosphere, so the questions that I ask myself are: does the present work unambiguously highlight reconnection being the majority component of the possibly competing heating events? and, do the diagnostics presented even permit such a discernment being made? In both situations I cannot answer in the affirmative.

There is a statement in the opening paragraph that wave energy [Alfvenic, Torsional, etc] is not sufficient to heat the active corona. I believe that this statement may be true but McIntosh et al follow with a statement that their assessment regarding wave energy is impeded by the patio-temporal resolution of the observations and there could be considerable amounts of "hidden energy". That's ok. There is a different issue however that, at the smallest spatial scales there is NO difference between the processes of wave dissipation and nano flare heating. The braiding, twisting, jostling of the field lines [which are invisible] induced by magneto-convection is really a wave process and the dissipation of induced small-angle differences in the magnetic field are "nanoflares." I refer the authors to two recent and popular reviews:

<http://adsabs.harvard.edu/abs/2012RSPTA.370.3193D> and

<http://adsabs.harvard.edu/abs/2012RSPTA.370.3217P>

My issues with the bulk of the manuscript can be broken into those two categories, 1) the modeling approach and 2) the diagnostics applied to. I will address these separately, as follows:

Modeling Methodology:

The model is static, there is no flux emergence - can the authors comment on the validity of excluding flux emergence as driving heating, surely the impact could/would be "nanoflares" too. I am to believe that the driving velocities are easily discernible from observations. That is likely not the case. The driving velocities must be subject to noise characteristics in the methodology applied. Can the "field lines" be resolved at BBSO/NST resolution, can they demonstrate - I'm highly skeptical that the resolution of HMI is extremely out of range. Are chromospheric diagnostics [imaging or otherwise] used to help clarify? This approach is a theoretical toy. Also, what is the statistical range of alpha parameters in the NLFF extrapolations and how do they vary locally and globally over the modeled field of view as well as over time. I am VERY concerned that the noise induced by forcing a NLFF solution is actually what you're seeing the impact of and not any physical transport. What is the velocity spectrum applied? Is it an observational one? There's simply no completeness here to establish the applicability of the method.

Diagnostic Methodology

- DEM is a blunt knife. It is a measure of density square weighted emission of plasmas at a given temperature. It is a function of density and temperature as the authors identify. But as used it can characterize the apparent distribution of temperatures of the emitting plasma in a pixel of code/observation from relatively broad (in temperature) transmission functions. It is not a unique method. The broadband transmission profiles are nowhere near as exact as the state-of-the-art approach of reproducing line intensities, widths and Doppler velocities that are required to back up claims made that go beyond the "well nanoflares must be important" presumption in the community.

- While qualitatively giving the appearance of reproducing the observations, depending on the colorable and range chosen, critical elements are missing indicating that the final temperature distribution in the active region modeled is not correct and the thermodynamics are incorrect - there are extensive regions of "moss" in the field of view. Moss lies in the transition region of hot, high pressure loops. What does the lack of moss being reproduced tell us about the modeling and/or diagnostic methodology applied?

Do you feel that the paper will influence thinking in the field?

> No. Too many simplistic approximations made in analysis and methodology.

We would also be grateful if you could comment on the appropriateness and validity of any statistical analysis, as well the ability of a researcher to reproduce the work, given the level of detail provided.

> See above.

Reviewer #2 (Remarks to the Author):

REFEREE REPORT

I consider the work of extreme interest not only for physics of the solar corona, but for stellar coronae in general and possibly also for other applications in magnetized plasmas.

The slipping reconnection is an important energy release mechanism known to be present in solar flares, both predicted theoretically and later confirmed observationally. The idea to apply the slipping reconnection mechanism to coronal heating and modeling of coronal emission is novel; as is the finding that the slipping velocity is structured on small spatial scales. The observational paper of Aulanier et al. (2007, *Sci* 318, 1588) and a short comment in Testa et al. (2013, *Astrophys. J.*, 770, 1) represent the only clues I am aware of that the slipping reconnection could be important in coronal physics as well.

The authors succeed in generating a coronal emission that both contains coronal loops and requires no fudge (filling) factor to scale the modeled emission to the observed one. Both these points are important, given that some of the previous attempts failed in either or both accounts.

However, the coronal emission synthesis employed ought to be improved upon, as there are a number of difficulties that must be addressed.

COMMENTS AND SUGGESTIONS

[1] The synthesis of temperature and density (from which the DEM(T) and AIA emission is calculated) is simplistic, and in turn may not be entirely realistic, since it

i] disregards the loop geometry as calculated from the NLFFF extrapolation of the HMI vector magnetograms; both assuming semi-circular geometry (Eq. M2) and no expansion of the magnetic flux tubes. The latter is not even commented upon.

ii] assumes equilibrium solutions, i.e., no time derivative in Eq. (M1). This perhaps could be justified by public unavailability of the 1D hydrodynamic codes. However, the manuscript finds that the value of R , representing a ratio of the heating scale-length to loop half-length $L/2$, is optimally chosen at $R=0.3$. This finding is problematic, since short heating scale-lengths could prevent equilibrium solutions, a well-known fact already noted by Serio et al. (1981, *Astrophys. J.*, 243, 288) and subsequently by Aschwanden et al. (2001, *Astrophys. J.*, 550, 1036).

The importance of geometric effects and their connection to departures from thermal equilibrium in coronal loops are discussed in works of Dudik et al. (2011, *Astron. Astrophys.*, 531, A115) and Mikic et al. (2013, *Astrophys. J.*, 773, 94). In particular, combinations of expanding cross-sections and short heating scale-lengths could produce thermal non-equilibrium in coronal loops; evidence of which has been detected in the solar corona (e.g., Froment et al. 2015, *Astrophys. J.*, 807, 158; Froment et al. 2017, *Astrophys. J.*, 835, 272; see also Mok et al. 2016; *Astrophys. J.*, 817, 15).

[2] Furthermore, stating that the loops are "in equilibrium" because "no significant flares occurred" is misleading, since observed loops are known to be cooling, i.e., temporally variable. After all, the authors cite one of the most important papers on observations of ubiquitous cooling of active region loops (Viall & Klimchuk 2012, *Astrophys. J.*, 753, 35).

Moreover, the authors find a reconnection (heating) interval τ_r to be "far greater than the cooling time" (page 4, line 93). This statement appears to be at odds with the equilibrium assumption.

[3] Regarding modeling of coronal emission, the authors cite only the paper of Schrijver et al. (2004, *Astrophys. J.*, 615, 512), while an entirety of newer literature on the subject is ignored. This is a large omission. An incomplete list some of the papers dealing with synthesis of coronal emission is provided below, and I request that authors cite at least several of these papers:

Mok et al. 2005, *Astrophys. J.*, 621, 1098
Mok et al. 2008, *Astrophys. J.*, 679, 161
Mok et al. 2016, *Astrophys. J.*, 817, 15
Warren & Winebarger 2006, *Astrophys. J.*, 645, 711
Warren & Winebarger 2007, *Astrophys. J.*, 666, 1245
Winebarger et al. 2008, *Astrophys. J.*, 676, 672
Lundquist et al. 2008, *Astrophys. J. Suppl. Ser.*, 179, 509
Lundquist et al. 2008, *Astrophys. J.*, 689, 138
Martinez-Sykora et al. 2011, *Astrophys. J.*, 743, 23
Dudik et al. 2011, *Astron. Astrophys.*, 531, A115
Peter & Bingert 2012, *Astron. Astrophys.*, 548, A1
Lionello et al. 2013, *Astrophys. J.*, 773, 134
Bourdin et al. 2013, *Astron. Astrophys.*, 555, A123
Bourdin et al. 2016, *Astron. Astrophys.*, 589, A86
Olluri et al. 2015, *Astrophys. J.*, 802, 5
Bradshaw & Viall 2016, *Astrophys. J.*, 821, 63

[4] The intensity histograms of the modeled emission are mentioned in the Abstract as well as Results and Discussion, but are not shown. Only a total χ^2 (summed over six filters) is shown in Fig. 2c. Omitting the intensity histograms precludes further quantitative comparison among the

observations and the model.

[5] The authors cite articles from The Astrophysical Journal (13) and Solar Physics (6), but only two articles from Astronomy & Astrophysics. The authors should consider whether there might be an unconscious bias.

[6] The reference #3 (Zou et al. 2017) does not seem to be appropriate on page 1, line 19. This paper concerns observations of an active region filament rather than a coronal structure.

[7] The reference #23 (Viall & Klimchuk 2012) does not seem to be appropriate on page 4, line 103. These authors do not build a model of active region corona.

[8] The CHIANTI 8.1 atomic database and software is not properly referenced, although it is essential to building the model of coronal emission. The proper references (see www.chiantidatabase.org/referencing.html) are

Dere et al. 1997, Astron. Astrophys. Suppl., 125, 149

Del Zanna et al. 2015, Astron. Astrophys., 582, A56

[9] Please note that the "latest" (page 8, line 204) 'hybrid' abundances Schmelz et al. (2012) are not necessarily the 'correct' coronal abundance values, as the FIP bias in corona may depend e.g. on age or individual structure. Using a different set of abundances (photospheric or coronal) would change the total radiative losses $\Lambda(T)$, by a factor of about 2, but will not influence the visualization process of AIA images itself, since in coronal conditions the AIA images are dominated only by Fe ions.

Jaroslav Dudik
referee

Reviewer #3 (Remarks to the Author):

Here is my review for the manuscript entitled

Observationally quantified reconnection providing a viable mechanism for active region coronal heating
Yang et al.

The manuscript addresses the relevant issue of seeking for observational support for the nanoflare scenario for coronal heating in active regions. The paper is very well written and the authors make a compelling case using a schematic theoretical model and the observational data at their disposal. However, I think a few words of caution should be added to the text, before I can recommend publication of this manuscript.

I agree with their back-of-the-envelope calculation to estimate the heating rate, and I understand that the authors do their best to come up with constraints for the free parameters in this calculation. However, for the benefit of the readers, I think the authors should stress that these are crude estimates at best. I am referring for instance to the diameter L for the flux elements being reconnected. Based on Big Bear data, 160 km would definitely be an upper limit rather than a typical value. Chances are that there is a whole range of values of L for the very many

reconnection events consistent with the nanoflare scenario. There is no reason to believe that the events that Big Bear managed to resolve just barely, happen to be typical dissipation events. Also, the authors assume that the heating time τ_r is much longer than typical cooling times, but that is based on the same assumption that 160 km is the typical size and not an upper limit. I urge the authors to take these considerations into account and warn the reader in this direction.

Responses to Reviewers' Comments

We thank the reviewers very much for a careful reading and constructive comments on the paper. Based on these comments, we have made a substantial revision of the paper. We hope this revision can meet the requirements and answer the questions by the reviewers. Our revisions in the paper are in cyan color. The replies to the reviewers' questions are listed as follows. With a recent discussion on the terms, we have changed the term 'slipping velocity' to 'non-ideal velocity'.

Reviewer 1,

Reviewer's general comments:

Q: *Unfortunately I do not find the work contained in the manuscript terribly novel. Also, I do not think that the outcomes are unique, largely as a result of the model-observation metric applied. As a result it is very difficult to assess if the rather obvious claim being made in the title and abstract is indeed the case. I will provide some examples below, but cannot see how anything other than a radically different approach could be used to back the claim made above.*

A: **Thanks for the comments. We would express our point of view on the novelty of the paper. We think that our results are novel in the following aspects. We consider the effect of 3D reconnection of magnetic fields in a novel way that the conjugate footpoints show a departure from the ideal plasma flow. And we do NOT use the velocity spectrum as the input to the model. We do not drive the system with an imposed motions with specified spectral properties, as the referee implies. Instead we use a novel form of heating expression which incorporates measured quantities and only a single free parameter, \mathcal{R} . The most important is that the parameter, \mathcal{L} , could be constrained by the high resolution observations instead of an artificial high resistivity, which can not be measured directly. Even the coronal model (we use the equilibrium state as the approximation of the plasma response to the heating) is simple, however, its predictions are remarkably similar to the observations.**

As shown in the main text, we calculate the non-ideal velocity by combining the techniques of DAVE4VM and the optimization method of the NLFFF. These techniques offer a reasonable way to infer the physical parameters from the observations, \mathbf{V} and \mathbf{B} , which are used to test our heating expression. The diagnostic methodology uses not only

the intensity histogram but also the DEM inversion technique. Although the DEM method is an ill-posed problem, if we vary the input data and try different codes as shown in Figure R5, we could still trust the region where the solution is stable. Considering that there are no other more accurate and practical methods at present, we think we have done our best in performing such an investigation. We have also estimated the errors by 50 Monte-Carlo realizations (Figure 5 in the main text). The result shows that our calculation is trustable with acceptable errors.

Q: *Are they novel and will they be of interest to others in the community and the wider field?*

I [opinion] think it is widely believed that magnetic reconnection at small-scales (nanoflares; as discussed originally by Parker) is important for the transport of mass and energy throughout the outer solar atmosphere, so the questions that I ask myself are: does the present work unambiguously highlight reconnection being the majority component of the possibly competing heating events? and, do the diagnostics presented even permit such a discernment being made? In both situations I cannot answer in the affirmative. There is a statement in the opening paragraph that wave energy [Alfvenic, Torsional, etc] is not sufficient to heat the active corona. I believe that this statement may be true but McIntosh et al follow with a statement that their assessment regarding wave energy is impeded by the patio-temporal resolution of the observations and there could be considerable amounts of ‘hidden energy’. That’s ok. There is a different issue however that, at the smallest spatial scales there is NO difference between the processes of wave dissipation and nano flare heating. The braiding, twisting, jostling of the field lines [which are invisible] induced by magneto-convection is really a wave process and the dissipation of induced small-angle differences in the magnetic field are ‘nanoflares’. I refer the authors to two recent and popular reviews: <http://adsabs.harvard.edu/abs/2012RSPTA.370.3193D> and <http://adsabs.harvard.edu/abs/2012RSPTA.370.3217P>

A: Thanks for the comments. Two main classes of heating mechanism have been proposed, waves and nano-flares. Here we focus only on the second possibility and agree with the reviewer that waves cannot be ruled out. In particular we are not assessing their relative importance. For a comparison the two review papers suggested by the reviewer are really worth citing. We have revised our statements on wave-based models and added a small discussion in the last part of the first paragraph in the Introduction section (Line 30–37 Page 2).

Comments on Modelling methodology:

Q: *The model is static, there is no flux emergence - can the authors comment on the validity of excluding flux emergence as driving heating, surely the impact could/would be nanoflares too.*

A: Our model is not totally static, since we use a time series of velocity field at the lower boundary and along with a 3D magnetic field to deduce the changing of the field line mapping and further the heating flux. We do assume an equilibrium loop in order to derive the temperature and density resulting from the inferred heat input. We have added some text in our revised manuscript in the section of Modelling the Corona to explain this approximation (Line 101–110 Page 4). There we point out that our approximation can, in fact, accommodate time-dependent heating at frequencies implied by observations. Nanoflares could be driven by flux emergence (vertical velocity) and/or shearing and twisting motions (horizontal velocity). In this paper, we only model the horizontal velocity. As the flux emergence will affect the field line mapping in the corona and the photospheric flow, in order to apply our model, we have carefully selected an active region in which no significant flux emergence or cancellation is occurring (see Fig R1). Some sentences have been added in the first paragraph of the new section of Application to Observations (Line 125–127 Page 5).

Figure R1– The variation of the magnetic flux of our active region. The period in our investigation is covered by the light gray region. The relative variation of the unsigned, positive, and negative fluxes during the two hours are 0.4%, 0.5%, 0.8%, respectively.

Q: *I am to believe that the driving velocities are easily discernible from observations. That is likely not the case. The driving velocities must be subject to noise characteristics in the methodology applied.*

A: Thanks for the comments. We agree that it is difficult to calculate the driving velocities, but we are here making the best attempt with state-of-the-art observations, namely from HMI. We agree that noise is a limitation and thus we have now attempted to estimate the level of errors in our computation of the heating (shown in Figure R2). Among all the features, we note that those locations with the largest values of the plasma velocity, non-ideal velocity and heating flux, have the smallest errors (Figure R2 a, b, and c). In those locations with magnetic field strength greater than 100 G, the relative errors of the plasma velocity and non-ideal velocity are less than 0.6 (Figure R2 d and e). Moreover, when the non-ideal velocity and heating flux increase, the relative errors decrease (Figure 2R g, h, and i). In addition, the non-ideal velocity is larger than the plasma velocity, but it does not matter, since it is only an apparent motion, not a physical motion. A figure of these errors has been added in the revised manuscript as Figure 5, and a discussion on the errors has been added in the first paragraph of the Discussion section (Line 155–160 Page 6).

Figure R2– The errors of the plasma velocity (a, d, and g), non-ideal velocity (b, e, and h), and heating flux (c, f, and i), coming from 50 Monte-Carlo realizations with a random error of 20 G in the bottom

vector magnetogram. The relative errors are estimated as the standard deviation over the mean value. (a)–(c) show the relative error distributions of the plasma velocity, the non-ideal velocity, and heating flux, respectively. (d)–(f) show the 2D histogram density distributions of the errors depending on the magnitude of the magnetic field. (g)–(i) show the 2D histogram density distributions of the errors depending on the mean values of the corresponding parameters.

Q: *Can the ‘field lines’ be resolved at BBSO/NST resolution, can they demonstrate - Im highly skeptical that the resolution of HMI is extremely out of range.*

A: We do not mean that the BBSO/NST can observe the ‘field lines’ associated with the small scale reconnection. The observed width of the magnetic strands might not directly infer the diameter of the reconnected tube. The observations do show, however, that some fine structures down to the scale of our inferred value of the parameter \mathcal{L} (Ji et al. 2012, Cirtain et al. 2013). We have added a notation on this point at the last part of the section of Energy Released by Magnetic Reconnection (Line 76–81 Page 3). We agree with the referee that the resolution of HMI itself is not enough to show the ‘field lines’ associated with the small scale reconnection. What we had obtained from the HMI data are the magnetic field line mapping and the bottom flow.

Q: *Are chromospheric diagnostics [imaging or otherwise] used to help clarify? This approach is a theoretical toy. Also, what is the statistical range of alpha parameters in the NLFF extrapolations and how do they vary locally and globally over the modeled field of view as well as over time. I am VERY concerned that the noise induced by forcing a NLFF solution is actually what youre seeing the impact of and not any physical transport. What is the velocity spectrum applied? Is it an observational one? Theres simply no completeness here to establish the applicability of the method.*

A: We have not attempted to use chromospheric diagnostics. For the region under study, we have no observations for chromospheric magnetic fields. Chromospheric images do not help much in determining the non-ideal velocity which our model needs. We have estimated the statistical range of the alpha parameter in the force-free calculation, as suggested by the reviewer. Its absolute values fall in the range of $0 - 3$ (10^{-1} Mm^{-1}) in the area where $|\mathbf{B}| > 50 \text{ G}$ and $0 - 10$ (10^{-1} Mm^{-1}) in the area where $|\mathbf{B}| < 50 \text{ G}$. Naturally, the lower the magnetic strength the larger the errors. The α distribution over time is shown in Figure R3, which

shows that during 2 hours, it changes very little. A Monte Carlo test has been conducted by adding random, normally distributed errors to the photospheric magnetogram, and repeating our analysis 50 times. The reduced plasma velocity, non-ideal velocity, heating, and the corresponding errors are shown in Figure R2. We are not imposing a velocity spectrum, but are deducing the footpoint velocities from observations, and then combining the magnetic field to measure the non-ideal velocity. However, as the reviewer requested, we show the plasma velocity spectrum and the non-ideal velocity spectrum in Figure. R4.

Figure R3– The α distribution from 16:58 to 19:01 UT.

Figure R4– The velocity spectrum of the plasma and non-ideal velocities. The blue-red lines denote the spectrum of the x and y components of the non-ideal velocities, and the solid-dash black lines denote the spectrum of the x and y components of the plasma velocities.

Comments on Diagnostic methodology:

Q: *-DEM is a blunt knife. It is a measure of density square weighted emission of plasmas at a given*

temperature. It is a function of density and temperature as the authors identify. But as used it can characterize the apparent distribution of temperatures of the emitting plasma in a pixel of code/observation from relatively broad (in temperature) transmission functions. it ohs not a unique method. The broadband transmission profiles are nowhere near as exact as the state-of-the art approach of reproducing line intensities, widths and Doppler velocities that are required to back up claims made that go beyond the ‘well nanoflares must be important’ presumption in the community.

A: We agree that the DEM method has limitations. In order to test the reliability of our method, we have compared several different methods for deriving DEMs (in particular the sparse DEM inversion from Cheung et al. 2015). The result shows a similar DEM profile, though in Cheung’s method the DEM is defined as $d[EM(\log(T))]/d[\log(T)]$. On the other hand, we also perform a DEM inversion using the synthesized EUV images predicted from our model (shown in Figure R5). Even though the result has larger errors, the peak in high temperatures is consistent with that inverted from the observation and that derived directly from the model. These tests suggest that our DEM method can yield reasonable results, though it have some limitations. We have added a discussion on the limitations in the Discussion section (Line 165–171 Page 6). We have also added an image of the comparison of DEM profiles constructed from the model, inverted from observations and from synthesized EUV images in the manuscript as Figure 6 (see also Figure R5).

Figure R5– Left, Comparison of DEM profiles constructed from the model (blue), inverted from observations (black) and from synthesized EUV images (Red). Right, The DEM profile inverted by using Cheung’s sparse method.

Q: -While qualitatively giving the appearance of reproducing the observations, depending on the colorable and range chosen, critical elements are missing indicating that the final temperature distribution in the active region modeled is not correct and the thermodynamics are incorrect—there are extensive regions of ‘moss’ in the field of view. Moss lies in the transition region of hot, high pressure loops. What does the lack of moss being reproduced tell us about the modeling and/or diagnostic methodology applied?

A: We agree that the comparison between the model and observations has some limitations. In fact, our model is capable of reproducing moss, and several observed moss regions also appear in our synthetic images. Two such cases are indicated by the red and blue circles (Figure R6). Though the emission along the loop departs from the observation, the main element can be identified. In the revised manuscript, we have added some numbers to label the corresponding structures between the synthesized EUV images and the observations in Figure 4. We should mention that the equilibrium state is only the first order approximation. In this research, we do not attempt to build up a coronal model combining the chromosphere and transition region entirely correct. The coronal model is used to check the heating expression. For the calculation to be practical, we chose the equilibrium loop model that enables us to do the optimal parameter searching.

Figure R6– The observed (left) and synthetic (right) AIA 171 Å images. The red and blue circles indicate the corresponding moss structures.

Reviewer 2,

Q: [1] The synthesis of temperature and density (from which the $DEM(T)$ and AIA emission is calculated) is simplistic, and in turn may not be entirely realistic, since it

i) disregards the loop geometry as calculated from the NLFFF extrapolation of the HMI vector

magnetograms; both assuming semi-circular geometry (Eq. M2) and no expansion of the magnetic flux tubes. The latter is not even commented upon.

ii] assumes equilibrium solutions, i.e., no time derivative in Eq. (M1). This perhaps could be justified by public unavailability of the 1D hydrodynamic codes. However, the manuscript finds that the value of R , representing a ratio of the heating scale-length to loop half-length $L/2$, is optimally chosen at $R=0.3$. This finding is problematic, since short heating scale-lengths could prevent equilibrium solutions, a well-known fact already noted by Serio et al. (1981, *Astrophys. J.*, 243, 288) and subsequently by Aschwanden et al. (2001, *Astrophys. J.*, 550, 1036).

The importance of geometric effects and their connection to departures from thermal equilibrium in coronal loops are discussed in works of Dudik et al. (2011, *Astron. Astrophys.*, 531, A115) and Mikic et al. (2013, *Astrophys. J.*, 773, 94). In particular, combinations of expanding cross-sections and short heating scale-lengths could produce thermal non-equilibrium in coronal loops; evidence of which has been detected in the solar corona (e.g., Froment et al. 2015, *Astrophys. J.*, 807, 158; Froment et al. 2017, *Astrophys. J.*, 835, 272; see also Mok et al. 2016; *Astrophys. J.*, 817, 15).

A: i] Thanks for the comments. We agree with the referee that our approach may not be ‘entirely realistic’. We have two reasons for not using the real geometry from the NLFFF.

- Since we populate every coronal point independently rather than superposing distinct loops, in our data we have $341 \times 501 \times 501$ points. If we use the real geometry of the field lines from the NLFFF, we would need to solve the equilibrium equations for 8×10^7 separate loops. This seems impractical at present, and would take an undue amount of time and computer resources.
- The way the assumed axis geometry enters our model is actually a rather subtle issue. Every volume element is populated independently using a single point taken from an equilibrium loop. We just use the physical parameters at this single point but not the whole loop; thus semi-circular loops are not evident in any of our model results. We use the true distance along the real (non-semi-circular) field line to determine which point along the equilibrium loop to draw the physical parameters from.

The majority of every loop is at high enough temperature that thermal conduction is extremely effective and the pressure scale height is comparable or larger than the loop’s full length. This means that our assumed geometry has very little effect on the precise values of density and temperature, which we find and use for the vast majority

of the coronal volume. The only place where this reasoning breaks down is near the feet where temperature is low.

The loop's feet will not be affected by the detailed shape we adopt for the entire axis, but they will be affected by the angle from vertical the axis makes at the footpoint: the inclination angle of the legs. We had assumed that the legs were perfectly vertical, which is admittedly a singular case. Under that assumption density and pressure fell off with the small length scale along the axis. In point of fact, the loops found in the NLFFF have a distribution of inclination angles, which will lead to a distribution of density scale distances: all systematically larger than ours. We have therefore chosen to quantify how the distribution of inclination angles will affect our results. We measure the inclination of the loop footpoint at the bottom and show them in Figure R7. We can see that the loops with a lower height usually have a large inclination angle, while the higher one is nearly perpendicular to the bottom (close to the semi-circular structure). And we make a comparison between the oblique loop and the semi-circular geometry with a typical heating value (Figure R8). Yes, it departs from the semi-circular loop with the inclination increases; however, the difference between them is rather small, that makes the semi-circular loop be a good approximation. On the other hand, we show the expansion factor calculated with the flux tube distributed uniformly at the bottom plane in Figure R9. It shows that the lower loops expand weakly while the higher loop expand largely. This is reasonable. Considering the loop solution, the most affected by the flux expansion is the transition region, where the temperature is relatively lower than the corona and we do not have a good detection from the DEM inversion technique with AIA data. The flux expansion effect in the corona can be converted to a change of the heating scale length (Aschwanden & Schrijver 2002). They give a relation between the modified heating scale length, s_H^Γ , and the original one, s_H , with the expansion factor,

$$s_H^\Gamma = \begin{cases} \frac{s_H}{\sqrt{1+(\Gamma-1)(s_H/L)}} & \text{for } s_H < L, \\ \frac{L}{\sqrt{\Gamma-1}} & \text{for } s_H > L, \end{cases}$$

where Γ is the expansion factor and L is the loop length. The value of parameter \mathcal{R} in our fitting results can be treated as the one that has been modified with the expansion factor. If we only consider the case $s_H < L$, then the parameter \mathcal{R} without expansion effect can be found in Figure R10. Thus the expansion effect here is only changing

the optimal value of parameter \mathcal{R} to a large one, e.g., $\mathcal{R}(\mathcal{R}^\Gamma = 0.3, \Gamma = 30) \approx 1$. Some discussions have been added in the 3rd and 4th paragraphs in the Discussion section (Line 178–186 Page 6–7).

Figure R7– The distribution of the inclination angle of the loop footprint. The top panel show the distribution of the inclination angle of total flux tube from each pixel (black curve), the loop height below 30 Mm (blue curve), the loop height over 30 Mm (green curve). The bottom panel shows the histogram density depending on the inclination angle and the loop top height.

Figure R8– The top left shows the structure of the oblique loop plane and the top right shows the loop footprint that is not perpendicular to the bottom. As an example, we calculate the equilibrium loop with a 200 Mm loop length and a typical heating value of $10^{-2} \text{ erg s}^{-1} \text{ cm}^{-3}$. The bottom left and bottom right show the temperature distribution of the corresponding loop structure with a variation of the angle θ .

Figure R9– Top panel shows the normalized PDF of the expansion factor and the bottom panel shows the histogram density depending on the loop top height and expansion factor.

Figure R10– The ratio of scale heating length (\mathcal{R}_H) if considering the expansion factor, as a function of the revised scale heating length and the expansion factor (Aschwanden & Schrijver 2002). The dash line indicates the revised ratio of heating scale length of $\mathcal{R}_H^\Gamma = 0.3$.

A: ii] As this research is mainly focusing on the novel heating expression with the non-ideal velocity, we opt to model the coronal response using the simplest tool: an equilibrium loop. Though in the real case, the loop will experience a dynamic process. Most loops are close to the equilibrium state for a long time period. The equilibrium loop would be a first order approximation of the corona. In this research, we do not attempt to build up an entire coronal model combing the chromosphere, transition region, and the dynamic process entirely correct. The coronal model here is used to check the heating expression by comparing some observables with observations. For computational practicability, we chose the equilibrium loop mode that enables us to do the optimal parameter searching. From the result, the heating rate inferred by our model with a reasonable parameter is large enough to power the corona, in contrast to the heating rate by nano-flares extrapolated from the occurrence distribution of larger flares that is not enough for the corona.

From Aschwanden et al. (2001), for the steady loop solution, the empirical limit of the heating scale length is $s_{H(Mm)} \geq \sqrt{L_{(Mm)}/2}$, which leads to a ratio $\mathcal{R} = s_H/(L/2) > 1/\sqrt{L_{(Mm)}/2}$. In our research, when the parameter lies in the region where there is no solution, we just remove it from the database. We can see from the synthetic EUV images that most of the loops are very large. In Figure R11, we show the χ^2 for each AIA band rather than the total Pearson χ^2 in the manuscript. We can see that the local minimum for AIA 131, 171, 193, and 211 Å bands is accompanied with a larger value of \mathcal{R} than that in the case of total χ^2 . The factor that causes the lower value of \mathcal{R} in the latter case is mainly from AIA 94 and 335 Å bands, since the total Pearson χ^2 is not normalized for each waveband. However, the locations of the local minimum of parameter \mathcal{L} for each band are very close to each other, which means that the heating expression works very well. The only problem comes from how we treat the plasma response to the heating. Nevertheless, at least in the first order approximation, the equilibrium assumption could give a reasonable result.

Figure R11– The Pearson χ^2 of the intensity histogram (shown in Figure 3 in the main text) for six AIA bands.

Q: [2] Furthermore, stating that the loops are ‘in equilibrium’ because ‘no significant flares occurred’ is misleading, since observed loops are known to be cooling, i.e., temporally variable. After all, the authors cite one of the most important papers on observations of ubiquitous cooling of active region loops (Viall & Klimchuk 2012, *Astrophys. J.*, 753, 35).

Moreover, the authors find a reconnection (heating) interval τ_r to be ‘far greater than the cooling time’ (page 4, line 93). This statement appears to be at odds with the equilibrium assumption.

A: We thank the referee for this comment. It seems that we had written this incorrectly. From our analysis, the nano-flare frequency is about $5 \text{ km s}^{-1} / 160 \text{ km} \sim 0.03 \text{ Hz}$. This would be a truly high-frequency nano-flare storm. This means that in our case nanoflares would appear as a steady continuous heating, as had been suggested in previous studies (Antiochos et al. 2003, Brooks et al. 2009). If the velocities in the coronal loop are mainly sub-sonic, the pressure can be balanced by the gravity along the loop, and the steady heating can be balanced by the thermal conduction and radiative cooling. In recognition of these facts we have modified the sentence to ‘*To justify our equilibrium assumption, we note that the reconnection event frequency for a typical non-ideal velocity is $5 \text{ km s}^{-1} / 160 \text{ km} = 0.03 \text{ Hz}$. Since this rate is high compared to radiative cooling rate, impulsive reconnection will have the effect of a steady input, known as a nanoflare storm. Though there are a lot of dynamic processes in the real solar corona, the equilibrium approximation is still a good one under conditions such as these described above.*’ (Line 105–110 Page 4). We have also changed the sentences on describing the active region to ‘*We expect this region to be well approximated by our equilibrium assumption because the magnetic flux variation was less than one percent during two hours and no obvious flares occurred during the time of our modelling*’ (Line 125–127 Page 5).

Q: [3] Regarding modeling of coronal emission, the authors cite only the paper of Schrijver et al. (2004, *Astrophys. J.*, 615, 512), while an entirety of newer literature on the subject is ignored. This is a large omission. An incomplete list some of the papers dealing with synthesis of coronal emission is provided below, and I request that authors cite at least several of these papers: Mok et al. 2005, *Astrophys. J.*, 621, 1098 Mok et al. 2008, *Astrophys. J.*, 679, 161 Mok et al. 2016, *Astrophys. J.*, 817, 15 Warren & Winebarger 2006, *Astrophys. J.*, 645, 711 Warren & Winebarger 2007, *Astrophys. J.*, 666, 1245 Winebarger et al. 2008, *Astrophys. J.*, 676, 672 Lundquist et al. 2008, *Astrophys. J. Suppl. Ser.*, 179, 509 Lundquist et al. 2008, *Astrophys. J.*, 689, 138 Martinez-Sykora et al. 2011, *Astrophys. J.*, 743, 23 Dudik et al. 2011, *Astron. Astrophys.*, 531, A115 Peter & Bingert 2012, *Astron. Astrophys.*, 548, A1 Lionello et al. 2013, *Astrophys. J.*, 773, 134 Bourdin et al. 2013, *Astron. Astrophys.*, 555, A123 Bourdin et al. 2016, *Astron. Astrophys.*, 589, A86 Olluri et al. 2015, *Astrophys. J.*, 802, 5 Bradshaw & Viall 2016, *Astrophys. J.*, 821, 63

A: Thanks for this suggestion. We have added these references in our revised version in the first paragraph of the Modelling the Corona section (Line 102 Page 4).

Q: [4] *The intensity histograms of the modeled emission are mentioned in the Abstract as well as Results and Discussion, but are not shown. Only a total χ^2 (summed over six filters) is shown in Figure 2c. Omitting the intensity histograms precludes further quantitative comparison among the observations and the model.*

A: We have added the intensity histogram for six AIA bands in the new Figure 3.

Q: [5] *The authors cite articles from The Astrophysical Journal (13) and Solar Physics (6), but only two articles from Astronomy & Astrophysics. The authors should consider whether there might be an unconscious bias.*

A: We find this to be an interesting bibliographic observation. We are aware of these and other many journals, and routinely read papers from all of them. Naturally, we choose to reference previous work solely based on its relevance to the work we are reporting, especially when we have made use of that work in our research. This may end up drawing from certain publications more than others, but that is purely the result of where we have found the relevant references, not where we have chosen to look. We do not intentionally bias our selection. Nor do we feel it is good practice to cite literature for reason other than its scientific relevance and usefulness. Nevertheless, in the revised version, we have cited more references from A&A.

Q: [6] *The reference #3 (Zou et al. 2017) does not seem to be appropriate on page 1, line 19. This paper concerns observations of an active region filament rather than a coronal structure.*

A: Originally, we use the reference paper to illustrate the high resolution of the observations from NST/BBSO, and some extremely fine fibers can be seen even though they are filament fibers. As the reviewer comments, we replace it with the paper of Ji et al. (2012), another paper on the ultrafine channels observed by the NST/BBSO.

Q: [7] *The reference #23 (Viall & Klimchuk 2012) does not seem to be appropriate on page 4, line 103. These authors do not build a model of active region corona.*

A: **We thank the reviewer for drawing our attention to this. It was an error. We had intended to cite Bradshaw & Viall (2016).**

Q: [8] *The CHIANTI 8.1 atomic database and software is not properly referenced, although it is essential to building the model of coronal emission. The proper references (see www.chiantidatabase.org/referencing.html) are Dere et al. 1997, *Astron. Astrophys. Suppl.*, 125, 149 Del Zanna et al. 2015, *Astron. Astrophys.*, 582, A56*

A: **We have added this reference in the revised manuscript (Line 358 Page 13).**

Q: [9] *Please note that the ‘latest’ (page 8, line 204) ‘hybrid’ abundances Schmelz et al. (2012) are not necessarily the ‘correct’ coronal abundance values, as the FIP bias in corona may depend e.g. on age or individual structure. Using a different set of abundances (photospheric or coronal) would change the total radiative losses $\Lambda(T)$, by a factor of about 2, but will not influence the visualization process of AIA images itself, since in coronal conditions the AIA images are dominated only by Fe ions.*

A: **We agree with the reviewer. We have removed the word ‘latest’ in the Equilibrium Loop of the Method section to avoid the misleading.**

Reviewer 3,

Q: *I agree with their back-of-the-envelope calculation to estimate the heating rate, and I understand that the authors do their best to come up with constrains for the free parameters in this calculation. However, for the benefit of the readers, I think the authors should stress that these are crude estimates at best. I am referring for instance to the diameter \mathcal{L} for the flux elements being reconnected. Based on Big Bear data, 160 km would definitely be an upper limit rather than a typical value. Chances are that there is a whole range of values of L for the very many reconnection events consistent with the nanoflare scenario. There is no reason to believe that the events that Big Bear managed to resolve just barely, happen to be typical dissipation events. Also, the authors assume that the heating time τ_r is much longer than typical cooling times, but that is based on the same assumption that 160 km is the typical size and not an upper*

limit. I urge the authors to take these considerations into account and warn the reader in this direction.

A: We thank the reviewer for this comment. We agree that 160 km is an upper limit rather than a typical value. We have emphasized that 160 km is an upper limit of the observed structures from BBSO/NST, we have also cited the observation from Hi-C (Line 80–81 Page 3). The reference paper on these structures have been replaced by Ji et al. (2012) and Cirtain et al. (2013), since the original paper is mainly focusing on the filament fiber. In particular, the relation between the observed width of the magnetic strands and the diameter of the reconnected tube is not clear. We should carefully deal with what the observations tell us. There should exist a range of the width of reconnected flux tubes but observations can not reveal such a range. The observations do show, however, some fine structures down to the scale of our value of the parameter \mathcal{L} (Ji et al. 2012, Cirtain et al. 2013). Thus, for simplicity, we take a value of 160 km for the parameter \mathcal{L} in the model and assume it is the same for all flux elements. We have added a notation on this point in the section of Energy Released by Magnetic Reconnection (Line 76–81 Page 3).

We are sorry that the sentence ‘*The heating time τ_r is much longer than typical cooling time*’ was written by a mistake. We have revised it to ‘*To justify our equilibrium assumption, we note that the reconnection event frequency for a typical non-ideal velocity is $5 \text{ km s}^{-1}/160 \text{ km} = 0.03 \text{ Hz}$, which is high compared to radiative cooling rate. Impulsive reconnection will have the effect of a steady input, known as a nanoflare storm.*’ The frequency of the nano-flare storm depends on the parameter \mathcal{L} ; thus the equilibrium assumption also relies on this parameter as the referee pointed out. We have addressed this issue in the section of Modelling the Corona (Line 101–110 Page 4). Furthermore, we have added a discussion on the effect of a lower frequency nano-flare storm in the Discussion section: ‘*There may, however, exist significantly larger events occurring at a significantly lower frequency, which would fall outside the equilibrium assumption. Larger, less frequent nano-flares are, in fact, known to produce locally high temperature and high density in the corona*’ (Line 175–177 Page 6).

REVIEWERS' COMMENTS:

Reviewer #1 (Remarks to the Author):

I am happy with the comprehensive revisions made to the manuscript.

Reviewer #2 (Remarks to the Author):

REFEREE REPORT
NCOMMS-17-16257A

The authors have carefully considered all issues raised and the manuscript has been revised to my satisfaction. I appreciate the amount of work that has been put into the revision.

Given that R is a global parameter of the model and not field-line dependent (as in Dudik et al. 2011), the results obtained with the equilibrium model are impressive indeed.

I also wish to thank the authors for pointing me to the interesting paper of Ji et al. (2012) that they now reference.

Nevertheless due to the number of points raised previously there still remains a few details that are listed below. These are not requests, but only minor suggestions.

1) The authors performed calculations regarding the effects of loop geometry on the resulting $T(s)$ profiles that are shown in Figure R8. For the sake of clarity however, the authors should explicitly state whether the hydrostatic effects in such inclined loops are taken into account when computing the corresponding density profiles.

2) Although likely, most of the loop cross-sectional expansion does not have to happen mostly at loop feet. The expansion of loop cross-section follows the $B(s)$ profile inversely, whatever the magnetic field strength profile along the loop.

3) I appreciate the intensity histograms shown in Figure 3 as a function of the parameter L ; however it might also be interesting to see the corresponding histograms calculated as a function of parameter R (e.g., for a chosen value of $L=160$ km).

4) The formulation on p. 6 line 165-166, "The DEM is a powerful diagnostic tool since it lies at the heart of all coronal emission." is not optimal and should be rephrased.

5) p. 7 line 193: "... to maintain basic coronal structures": Please rephrase as "...to maintain an active region corona remarkably similar to the observed one."

My best regards,
Jaroslav Dudik
Reviewer #2

Reviewer #3 (Remarks to the Author):

I am satisfied with the clarifications introduced by the authors to address my points. Therefore, I have no further objections and I am inclined to recommend publication.

RESPONSE TO REVIEWERS' COMMENTS:**Reviewer #1:**

I am happy with the comprehensive revisions made to the manuscript.

A: We thank the reviewer for the evaluation.

Reviewer #2:

The authors have carefully considered all issues raised and the manuscript has been revised to my satisfaction. I appreciate the amount of work that has been put into the revision. Given that R is a global parameter of the model and not field-line dependent (as in Dudik et al. 2011), the results obtained with the equilibrium model are impressive indeed. I also wish to thank the authors for pointing me to the interesting paper of Ji et al. (2012) that they now reference. Nevertheless due to the number of points raised previously there still remains a few details that are listed below. These are not requests, but only minor suggestions.

A: We thank the reviewer for the evaluation and comments.

Suggestion 1:

The authors performed calculations regarding the effects of loop geometry on the resulting $T(s)$ profiles that are shown in Figure R8. For the sake of clarity however, the authors should explicitly state whether the hydrostatic effects in such inclined loops are taken into account when computing the corresponding density profiles.

A: We have added a sentence at the end of the 2nd paragraph of the Methods section to indicate this point.

Suggestion 2:

Although likely, most of the loop cross-sectional expansion does not have to happen mostly at loop feet. The expansion of loop cross-section follows the $B(s)$ profile inversely, whatever the magnetic field strength profile along the loop.

A: We have added two sentences at the beginning of the 4th paragraph of the Discussion section to indicate this point.

Suggestion 3:

I appreciate the intensity histograms shown in Figure 3 as a function of the parameter L ; however it might also be interesting to see the corresponding histograms calculated as a function of parameter R (e.g., for a chosen value of $\mathcal{L}=160$ km).

A: We have added two figures (Supplementary Figures 1 and 2) to show the intensity histograms as a function of parameter \mathcal{R} .

Suggestion 4:

The formulation on p. 6 line 165-166, ‘The DEM is a powerful diagnostic tool since it lies at the heart of all coronal emission.’ is not optimal and should be rephrased.

A: We have rephrased this sentence as ‘The DEM is a promising diagnostic tool when we analyze the multi-wavelength coronal emissions.’

Suggestion 5:

p. 7 line 193: ‘... to maintain basic coronal structures’: Please rephrase as ‘...to maintain an active region corona remarkably similar to the observed one.’

A: We have rephrased this sentence as suggested by the referee.

Reviewer #3:

I am satisfied with the clarifications introduced by the authors to address my points. Therefore, I have no further objections and I am inclined to recommend publication.

A: We thank the reviewer for the evaluation.